# ABCDP: Approximate Bayesian Computation with Differential Privacy

**DOI:** 10.3390/e23080961

**Published:** 2021-07-27

**Authors:** Mijung Park, Margarita Vinaroz, Wittawat Jitkrittum

**Affiliations:** 1Computer Science, University of British Columbia, Vancouver, BC V6T 1Z4, Canada; 2Max Planck Institute for Intelligent Systems, 72076 Tübingen, Germany; mvinaroz@tuebingen.mpg.de; 3Department of Computer Science, University of Tübingen, 72076 Tübingen, Germany; 4Google Research, 80636 Munich, Germany; wittawat@google.com

**Keywords:** approximate Bayesian computation (ABC), differential privacy (DP), sparse vector technique (SVT)

## Abstract

We developed a novel approximate Bayesian computation (ABC) framework, *ABCDP*, which produces differentially private (DP) and approximate posterior samples. Our framework takes advantage of the sparse vector technique (SVT), widely studied in the differential privacy literature. SVT incurs the privacy cost only when a condition (whether a quantity of interest is above/below a threshold) is met. If the condition is sparsely met during the repeated queries, SVT can drastically reduce the cumulative privacy loss, unlike the usual case where every query incurs the privacy loss. In ABC, the quantity of interest is the distance between observed and simulated data, and only when the distance is below a threshold can we take the corresponding prior sample as a posterior sample. Hence, applying SVT to ABC is an organic way to transform an ABC algorithm to a privacy-preserving variant with minimal modification, but yields the posterior samples with a high privacy level. We theoretically analyzed the interplay between the noise added for privacy and the accuracy of the posterior samples. We apply ABCDP to several data simulators and show the efficacy of the proposed framework.

## 1. Introduction

Approximate Bayesian computation (ABC) aims to identify the posterior distribution over simulator parameters. The posterior distribution is of interest as it provides the mechanistic understanding of the stochastic procedure that directly generates data in many areas such as climate and weather, ecology, cosmology, and bioinformatics [1,2,3,4].

Under these complex models, directly evaluating the likelihood of data is often intractable given the parameters. ABC resorts to an approximation of the likelihood function using simulated data that are *similar* to the actual observations.

In the simplest form of ABC called *rejection ABC* [5], we proceed by sampling multiple model parameters from a prior distribution π: $\theta_{1},\theta_{2},\;\ldots ∼\pi$. For each $\theta_{t}$, a pseudo dataset $Y_{t}$ is generated from a simulator (the forward sampler associated with the intractable likelihood P(y|θ)). The parameter $\theta_{t}$ for which the generated $Y_{t}$ are similar to the observed $Y^{*}$, as decided by $\rho \left(Y_{t},Y^{*}\right)<\epsilon_{abc}$, is accepted. Here, ρ is a notion of distance, for instance, L2 distance between $Y_{t}$ and $Y^{*}$ in terms of a pre-chosen summary statistic. Whether the distance is small or large is determined by $\epsilon_{abc}$, a *similarity threshold*. The result is samples ${\left\{\theta_{t}\right\}}_{t=1}^{M}$ from a distribution, ${\tilde{P}}_{\epsilon }\left(\theta |Y^{*}\right)\propto \pi \left(\theta \right){\tilde{P}}_{\epsilon }\left(Y^{*}|\theta \right)$, where ${\tilde{P}}_{\epsilon }\left(Y^{*}|\theta \right)=\int_{B_{\epsilon }\left(Y^{*}\right)}P\left(Y|\theta \right)dY$ and $B_{\epsilon }\left(Y^{*}\right)=\left\{Y\;:\;\rho \left(Y,Y^{*}\right)<\epsilon_{abc}\right\}$. As the likelihood computation is approximate, so is the posterior distribution. Hence, this framework is named by *approximate* Bayesian computation, as we do not compute the likelihood of data explicitly.

Most ABC algorithms evaluate the data similarity in terms of summary statistics computed by an aggregation of individual datapoints [6,7,8,9,10,11]. However, this seemingly innocuous step of similarity check could impose a privacy threat, as aggregated statistics could still reveal an individual’s participation to the dataset with the help of combining other publicly available datasets (see [12,13]). In addition, in some studies, the actual observations are privacy-sensitive in nature, e.g., Genotype data for estimating tuberculosis transmission parameters [14]. Hence, it is necessary to privatize the step of similarity check in ABC algorithms.

In this light, we introduce an ABC framework that obeys the notion of *differential privacy*. The differential privacy definition provides a way to quantify the amount of information that the distance computed on the privacy-sensitive data contains, whether or not a single individual’s data are included (or modified) in the data [15]. Differential privacy also provides rigorous privacy guarantees in the presence of *arbitrary side information* such as similar public data.

A common form of applying DP to an algorithm is by adding noise to outputs of the algorithm, called *output perturbation *[16]. In the case of ABC, we found that *adding noise to the distance* computed on the real observations and pseudo-data suffices for the privacy guarantee of the resulting posterior samples. However, if we choose to simply add noise to the distance in every ABC inference step, this DP-ABC inference imposes an additional challenge due to the *repeated* use of the real observations. The *composition* property of differential privacy states that the privacy level degrades over the repeated use of data. To overcome this challenge, we adopt the *sparse vector technique* (SVT) [17], and apply it to the rejection ABC paradigm. The SVT outputs *noisy* answers of whether or not a stream of queries is above a certain threshold, where privacy cost incurs only when the SVT outputs at most *c* “above threshold” answers. This is a significant saving in privacy cost, as arbitrarily many “below threshold” answers are privacy cost free.

We name our framework, which combines ABC with SVT, as  *ABCDP* (approximate Bayesian computation with differential privacy). Under ABCDP, we theoretically analyze the effect of noise added to the distance in the resulting posterior samples and the subsequent posterior integrals. Putting together, we summarize our main contributions:We provide a novel ABC framework, ABCDP, which combines the *sparse vector technique* (SVT) [17] with the rejection ABC paradigm. The resulting ABCDP framework can improve the trade-off between the privacy and accuracy of the posterior samples, as the privacy cost under ABCDP is a function of the number of *accepted* posterior samples only.We theoretically analyze ABCDP by focusing on the effect of noisy posterior samples in terms of two quantities. The first quantity provides the probability of an output of ABCDP being different from that of ABC at any given time during inference. The second quantity provides the convergence rate, i.e., how fast the posterior integral using ABCDP’s noisy samples’ approaches that using non-private ABC’s samples. We write both quantities as a function of added noise for privacy to better understand the characteristics of ABCDP.We validate our theory in the experiments using several simulators. The results of these experiments are consistent with our theoretical findings on the flip probability and the average error induced by the noise addition for privacy.

Unlike other existing ABC frameworks that typically rely on a pre-specified set of summary statistics, we use a kernel-based distance metric called *maximum mean discrepancy* following K2-ABC [18] to eliminate the necessity of pre-selecting a summary statistic. Using a kernel for measuring similarity between two empirical distributions was also proposed in K-ABC [19]. K-ABC formulates ABC as a problem of estimating a conditional mean embedding operator mapping (induced by a kernel) from summary statistics to corresponding parameters. However, unlike our algorithm, K-ABC still relies on a particular choice of summary statistics. In addition, K-ABC is a soft-thresholding ABC algorithm, while ours is a rejection-ABC algorithm.

To avoid the necessity of pre-selecting summary statistics, one could resort to methods that automatically or semi-automatically learn the best summary statistics given in a dataset, and use the learned summary statistics in our ABCDP framework. An example is semi-auto ABC [6], where the authors suggest using the posterior mean of the parameters as a summary statistic. Another example is the indirect-score ABC [20], where the authors suggest using an auxiliary model which determines a score vector as a summary statistic. However, the posterior mean of the parameters in semi-auto ABC as well as the parameters of the auxiliary model in indirect-score ABC need to be estimated. The estimation step can incur a further privacy loss if the real data need to be used for estimating them. Our ABCDP framework does not involve such an estimation step and is more economical in terms of privacy budget to be spent than semi-auto ABC and indirect-score ABC.

## 2. Background

We start by describing relevant background information.

### 2.1. Approximate Bayesian Computation

Given a set $Y^{*}$ containing observations, **rejection ABC** [5] yields samples from an approximate posterior distribution by repeating the following three steps:(1)$$\begin{array}{cc} \theta & ∼\pi (\theta ), \end{array}$$(2)$$\begin{array}{cc} Y=\{y_{1},y_{2},\ldots \} & ∼P(y|\theta ), \end{array}$$(3)$$\begin{array}{cc} P_{\epsilon_{abc}}\left(\theta |Y^{*}\right) & ∼P_{\epsilon_{abc}}\left(Y^{*}|\theta \right)\pi \left(\theta \right), \end{array}$$
where the pseudo dataset *Y* is compared with the observations $Y^{*}$ via:(4)$$\begin{array}{cc} P_{\epsilon_{abc}}\left(Y^{*}|\theta \right) & =\int_{B_{\epsilon_{abc}}\left(Y^{*}\right)}P\left(Y|\theta \right)dY,\\ B_{\epsilon_{abc}}\left(Y^{*}\right) & =\{Y|\rho \left(Y,Y^{*}\right)\leq \epsilon_{abc}\}, \end{array}$$
where ρ is a divergence measure between two datasets. Any distance metric can be used for ρ. For instance, one can use the L2 distance under two datasets in terms of a pre-chosen set of summary statistics, i.e., $\rho \left(Y,Y^{*}\right)=D\left(S\left(Y\right),S\left(Y^{*}\right)\right)$, with an L2 distance measure *D* on the statistics computed by *S*.

A more statistically sound choice for ρ would be *maximum mean discrepancy* (MMD, [21]) as used in [18]. Unlike a pre-chosen finite dimensional summary statistic typically used in ABC, MMD compares two distributions in terms of all the possible moments of the random variables described by the two distributions. Hence, ABC frameworks using the MMD metric such as [18] can avoid the problem of non-sufficiency of a chosen summary statistic that may occur in many ABC methods. For this reason, in this paper, we demonstrate our algorithm using the MMD metric. However, other metrics can be used as we illustrated in our experiments.

#### Maximum Mean Discrepancy

Assume that the data Y⊂X and let k:X×X be a positive definite kernel. MMD between two distributions P,Q is defined as
(5)$$\begin{array}{c} {\mathrm{MMD}}^{2}\left(P,Q\right)=E_{x,x^{\prime }∼P}k\left(x,x^{\prime }\right)+E_{y,y^{\prime }∼Q}k\left(y,y^{\prime }\right)-2E_{x∼P}E_{y∼Q}k\left(x,y\right). \end{array}$$ By following the convention in kernel literature, we call ${\mathrm{MMD}}^{2}$ simply  MMD.

The Moore–Aronszajn theorem states that there is a unique Hilbert space H on which *k* defines an inner product. As a result, there exists a feature map ϕ:X→H such that $k\left(x,y\right)={\langle \phi (x),\phi (y)\rangle }_{H}$, where ${\langle \cdot ,\cdot \rangle }_{H}=\langle \cdot ,\cdot \rangle$ denotes the inner product on H. The MMD in (Equation 5) can be written as
$$\begin{array}{cc} {\mathrm{MMD}}^{2}\left(P,Q\right) & =∥E_{x∼P}\left[\phi \left(x\right)\right]-E_{y∼Q}\left[\phi \left(y\right)\right]{∥}_{H}^{2}, \end{array}$$
where $E_{x∼P}\left[\phi \left(x\right)\right]\in H$ is known as the (kernel) mean embedding of *P*, and exists if $E_{x∼P}\sqrt{k(x,x)}<\infty$ [22]. The MMD can be interpreted as the distance between the mean embeddings of the two distributions. If *k* is a characteristic kernel [23], then $P\mapsto E_{x∼P}\left[\phi \left(x\right)\right]$ is injective, implying that MMD(P,Q)=0, if and only if P=Q. When P,Q are observed through samples $X_{m}={\left\{x_{i}\right\}}_{i=1}^{m}∼P$ and $Y_{n}={\left\{y_{i}\right\}}_{i=1}^{n}∼Q$, MMD can be estimated by empirical averages [21] (Equation (Equation 3)): ${\hat{\mathrm{MMD}}}^{2}\left(X_{m},Y_{n}\right)=\frac{1}{m^{2}}\sum_{i,j=1}^{m}k\left(x_{i},x_{j}\right)+\frac{1}{n^{2}}\sum_{i,j=1}^{n}k\left(y_{i},y_{j}\right)-\frac{2}{mn}\sum_{i=1}^{m}\sum_{j=1}^{n}k\left(x_{i},y_{j}\right).$ When applied in the ABC setting, one input to $\hat{\mathrm{MMD}}$ is the observed dataset $Y^{*}$ and the other input is a pseudo dataset $Y_{t}∼p\left(\cdot |\theta_{t}\right)$ generated by the simulator given $\theta_{t}∼\pi \left(\theta \right)$.

### 2.2. Differential Privacy

An output from an algorithm that takes in sensitive data as input will naturally contain some information of the sensitive data D. The goal of differential privacy is to augment such an algorithm so that useful information about the population is retained, while sensitive information such as an individual’s participation in the dataset cannot be learned [17]. A common way to achieve these two seemingly paradoxical goals is by deliberately injecting a controlled level of random noise to the to-be-released quantity. The modified procedure, known as a DP mechanism, now gives a stochastic output due to the injected noise. In the DP framework, a higher level of noise provides stronger privacy guarantee at the expense of less accurate population-level information that can be derived from the released quantity. Less noise added to the output thus reveals more about an individual’s presence in the dataset.

More formally, given a mechanism M (a *mechanism* takes a dataset as input and produces stochastic outputs) and neighboring datasets D, $D^{\prime }$ differing by a single entry (either by replacing one’s datapoint with another, or by adding/removing a datapoint to/from D), the *privacy loss* of an outcome *o* is defined by
(6)$$L^{\left(o\right)}=\log\frac{P(M(D)=o)}{P(M\left(D^{\prime }\right)=o)}.$$ The mechanism M is called ϵ-DP if and only if $|L^{\left(o\right)}|\leq \epsilon$, for all possible outcomes *o* and for all possible neighboring datasets $D,D^{\prime }$. The definition states that a single individual’s participation in the data does not change the output probabilities by much; this limits the amount of information that the algorithm reveals about any one individual. A weaker or an *approximate* version of the above notion is (ϵ,δ)-DP: M is (ϵ,δ)-DP if $|L^{\left(o\right)}|\leq \epsilon$, with probability 1−δ, where δ is often called a failure probability which quantifies how often the DP guarantee of the mechanism fails.

Output perturbation is a commonly used DP mechanism to ensure the outputs of an algorithm to be differentially private. Suppose a deterministic function $h:D\mapsto R^{p}$ computed on sensitive data D outputs a *p*-dimensional vector quantity. In order to make *h* private, we can add noise to the output of *h*, where the level of noise is calibrated to the *global sensitivity* [24], $\Delta_{h}$, defined by the maximum difference in terms of some norm $||h\left(D\right)-h\left(D^{\prime }\right)\left|\right|$ for neighboring D and $D^{\prime }$ (i.e., differ by one data sample).

There are two important properties of differential privacy. First, the *post-processing invariance* property [24] tells us that the composition of any arbitrary data-independent mapping with an (ϵ,δ)-DP algorithm is also (ϵ,δ)-DP. Second, the *composability* theorem [24] states that the strength of privacy guarantee degrades with the repeated use of DP-algorithms. Formally, given an $\epsilon_{1}$-DP mechanism $M_{1}$ and an $\epsilon_{2}$-DP mechanism $M_{2}$, the mechanism $M\left(D\right):=(M_{1}\left(D\right),M_{2}\left(D\right))$ is $\left(\epsilon_{1}+\epsilon_{2}\right)$-DP. This composition is often-called *linear* composition, under which the total privacy loss linearly increases with the number of repeated use of DP-algorithms. The *strong* composition [17] [Theorem 3.20] improves the linear composition, while the resulting DP guarantee becomes weaker (i.e., approximate (ϵ,δ)-DP). Recently, more refined methods further improve the privacy loss (e.g., [25]).

### 2.3. AboveThreshold and Sparse Vector Technique

Among the DP mechanisms, we will utilize *AboveThreshold* and *sparse vector technique* (SVT) [17] to make the rejection ABC algorithm differentially private. AboveThreshold outputs 1 when a query value exceeds a pre-defined threshold, or 0 otherwise. This resembles rejection ABC where the output is 1 when the distance is less than a chosen threshold. To ensure the output is differentially private, AboveThreshold adds noise to both the threshold and the query value. We take the same route as AboveThreshold to make our ABCDP outputs differentially private. Sparse vector technique (SVT) consists of *c* calls to AboveThreshold, where *c* in our case determines how many posterior samples ABCDP releases.

Before presenting our ABCDP framework, we first describe the privacy setup we consider in this paper.

## 3. Problem Formulation

We assume a *data owner* who owns sensitive data $Y^{*}$ and is willing to contribute to the posterior inference.

We also assume a *modeler* who aims to learn the posterior distribution of the parameters of a simulator. Our ABCDP algorithm proceeds with the two steps:*Non-private step:* The modeler draws a parameter sample $\theta_{t}∼\pi \left(\theta \right)$; then generates a pseudo-dataset $Y_{t}$, where $Y_{t}∼P\left(y|\theta_{t}\right)$ for t=1,⋯,T for a large *T*. We assume these parameter-pseudo-data pairs ${\left\{\theta_{t},Y_{t}\right\}}_{t=1}^{T}$ are publicly available (even to an adversary).*Private step:* the data owner takes the whole sequence of parameter-pseudo-data pairs ${\left\{\left(\theta_{t},Y_{t}\right)\right\}}_{t=1}^{T}$ and runs our ABCDP algorithm in order to output a set of *differentially private* binary indicators determining whether or not to accept each $\theta_{t}$.

Note that *T* is the maximum number of parameter-pseudo-data pairs that are publicly available. We will run our algorithm for *T* steps, while our algorithm can terminate as soon as we output the *c* number of accepted posterior samples. So, generally, c≪T. The details are then introduced.

## 4. ABCDP

Recall that the only place where the real data $Y^{*}$ appear in the ABC algorithm is when we judge whether the simulated data are similar to the real data, i.e., as in (Equation 4). Our method hence adds noise to this step. In order to take advantage of the privacy analysis of SVT, we also add noise to the ABC threshold and to the ABC distance. Consequently, we introduce two perturbation steps.

Before we introduce them, we describe the global sensitivity of the distance, as this quantity tunes the amount of noise we will add in the two perturbation steps. For $\rho \left(Y^{*},Y\right)=\hat{\mathrm{MMD}}\left(Y^{*},Y\right)$ with a bounded kernel, then the sensitivity of the distance is $\Delta_{\rho }=O\left(1/N\right)$ as shown in Lemma 1.

**Lemma** **1.**($\Delta_{\rho }=O\left(1/N\right)$ for MMD). *Assume that*
$Y^{*}$
*and each pseudo dataset*
$Y_{t}$
*are of the same cardinality N. Set*
$\rho \left(Y^{*},Y\right)=\hat{\mathrm{MMD}}\left(Y^{*},Y\right)$
*with a kernel k bounded by*
$B_{k}>0$*, i.e.,* $\sup_{x,y\in X}k\left(x,y\right)\leq B_{k}<\infty$*. Then:*$$\begin{array}{c} \underset{(Y^{*},Y^{{*}^{\prime }}),Y}{\sup}\left|\rho \left(Y^{*},Y\right)-\rho \left(Y^{{*}^{\prime }},Y\right)\right|\leq \Delta_{\rho }:=\frac{2}{N}\sqrt{B_{k}} \end{array}$$
*and*
$\sup_{Y^{*},Y}\rho \left(Y^{*},Y\right)\leq 2\sqrt{B_{k}}.$


A proof is given in Appendix B. For $\rho =\hat{\mathrm{MMD}}$ using a Gaussian kernel, $k\left(x,y\right)=\exp\left(-\frac{{∥x-y∥}^{2}}{2l^{2}}\right)$ where l>0 is the bandwidth of the kernel, $B_{k}=1$ for any l>0.

Now, we introduce the two perturbation steps used in our algorithm summarized in Algorithm 1.
**Algorithm 1** Proposed *c*-sample ABCDP**Require:** Observations $Y^{*}$, Number of accepted posterior sample size *c*, privacy tolerance $\epsilon_{total}$, ABC threshold $\epsilon_{abc}$, distance ρ, and parameter-pseudo-data pairs ${\left\{\left(\theta_{t},Y_{t}\right)\right\}}_{t=1}^{T}$, and option RESAMPLE.  **Ensure:**$\epsilon_{total}$-DP indicators ${\left\{{\tilde{\tau }}_{t}\right\}}_{t=1}^{T}$ for corresponding samples ${\left\{\theta_{t}\right\}}_{t=1}^{T}$
1:Calculate the noise scale *b* by Theorem 1.2:Privatize ABC threshold: ${\hat{\epsilon }}_{abc}=\epsilon_{abc}+m_{t}$ via (Equation 7)3:Set count=04:**for**t=1,…,T**do**5: Privatize distance: ${\hat{\rho }}_{t}=\rho \left(Y^{*},Y_{t}\right)+\nu_{t}$ via (Equation 8)  6: **if**
${\hat{\rho }}_{t}\leq {\hat{\epsilon }}_{abc}$
**then**7:  Output ${\tilde{\tau }}_{t}=1$8:  count = count+19:  **if** RESAMPLE **then**10:   ${\hat{\epsilon }}_{abc}=\epsilon_{abc}+m_{t}$ via (Equation 7)11:  **end if**12: **else**13:  Output ${\tilde{\tau }}_{t}=0$14: **end if**15: **if** count ≥ c **then**16:  Break the loop17: **end if**18:**end for**


*Step 1: Noise for privatizing the ABC threshold.*(7)$$\begin{array}{cc} {\hat{\epsilon }}_{abc} & =\epsilon_{abc}+m_{t} \end{array}$$
where $m_{t}∼Lap\left(b\right)$, i.e., drawn from the zero-mean Laplace distribution with a scale parameter *b*.

*Step 2: Noise for privatizing the distance.*(8)$$\begin{array}{cc} {\hat{\rho }}_{t} & =\rho \left(Y^{*},Y_{t}\right)+\nu_{t} \end{array}$$
where $\nu_{t}∼Lap\left(2b\right)$.

Due to these perturbations, Algorithm 1 runs with the privatized threshold and distance. We can choose to perturb the threshold only once, or every time we output 1 by setting RESAMPLE to false or true. After outputting *c* number of 1’s, the algorithm is terminated. How do we calculate the resulting privacy loss under the different options we choose?

We formally state the relationship between the noise scale and the final privacy loss $\epsilon_{tot}$ for the Laplace noise in Theorem 1.

**Theorem** **1.**
*(Algorithm 1 is*
$\epsilon_{total}$
*-DP) thmmrejdp For any neighboring datasets*
$Y^{*},Y^{{*}^{\prime }}$
*of size N and any dataset Y, assume that*
ρ
*is such that*
$0<\sup_{(Y^{*},Y^{{*}^{\prime }}),Y}\left|\rho \left(Y^{*},Y\right)-\rho \left(Y^{{*}^{\prime }},Y\right)\right|\leq \Delta_{\rho }<\infty$
*. Algorithm 1 is*
$\epsilon_{total}$
*-DP, where:*
(9)$$\begin{array}{c} \epsilon_{total}=\left\{\begin{array}{cc} \frac{\left(c+1\right)\Delta_{\rho }}{b} & if\;RESAMPLE\;is\;False,\\ \frac{2c\Delta_{\rho }}{b} & if\;RESAMPLE\;is\;True. \end{array}\right. \end{array}$$


A proof is given in Appendix A. The proof uses linear composition, i.e., the privacy level linearly degrading with *c*. However, using the strong composition or more advanced compositions can reduce the resulting privacy loss, while these compositions turn pure-DP into a weaker, approximate-DP. In this paper, we focus on the pure-DP. For the case of RESAMPLE = True, the proof directly follows the proof of the standard SVT algorithm using the linear composition method [17], with an exception that we utilize the quantity representing the minimum noisy value of any query evaluated on $Y^{*}$, as opposed to the maximum utilized in SVT. For the case of RESAMPLE= False, the proof follows the proof of Algorithm 1 in [26].

Note that the DP analysis in Theorem 1. holds for other types of distance metrics and not limited to only MMD, as long as there is a bounded sensitivity $\Delta_{\rho }$ under the chosen metric. When there is no bounded sensitivity, one could impose a clipping bound *C* to the distance by taking the distance from $\min[\rho \left(Y_{t},Y^{*}\right),C]$, such that the resulting distance between any pseudo data $Y_{t}$ and $Y^{{*}^{\prime }}$ with modifying one datapoint in $Y^{*}$ cannot exceed that clipping bound. In fact, we use this trick in our experiments when there is no bounded sensitivity.

### 4.1. Effect of Noise Added to ABC

Here, we would like to analyze the effect of noise added to ABC. In particular, we are interested in analyzing the probability that the output of ABCDP differs from that of ABC: $P[{\tilde{\tau }}_{t}\neq \tau_{t}|\tau_{t}]$ at any given time *t*. To compute this probability, we first compute the probability density function (PDF) of the random variables $m_{t}-\nu_{t}$ in the following Lemma.

**Lemma** **2.**
*Recall $m_{t}∼Lap\left(b\right)$, $\nu_{t}∼Lap\left(2b\right)$. The subtraction of these yields another random variable $Z=m_{t}-\nu_{t}$, where the PDF of Z is given by*
(10)$$\begin{array}{cc} f_{Z}\left(z\right) & =\frac{1}{6b}\left[2\exp\left(-\frac{\left|z\right|}{2b}\right)-\exp\left(-\frac{\left|z\right|}{b}\right)\right]. \end{array}$$
*Furthermore, for a≥0, $G_{b}\left(a\right):=\int_{a}^{\infty }f_{Z}\left(z\right)\;dz=\frac{1}{6}\left[4\exp\left(-\frac{a}{2b}\right)-\exp\left(-\frac{a}{b}\right)\right]$, and the CDF of Z is given by $F_{Z}\left(a\right)=H\left[a\right]+\left(1-2H\left[a\right]\right)G_{b}\left(|a|\right)$ where H[a] is the Heaviside step function.*


See Appendix C for the proof. Using this PDF, we now provide the following proposition:

**Proposition** **1.**
*Denote the output of Algorithm 1 at time t by ${\tilde{\tau }}_{t}\in \left\{0,1\right\}$ and the output of ABC by $\tau_{t}\in \left\{0,1\right\}$. The flip probability, the probability that the outputs of ABCDP and ABC differ given the output of ABC, is given by $P\left[{\tilde{\tau }}_{t}\neq \tau_{t}|\tau_{t}\right]=G_{b}\left(\right|\rho_{t}-\epsilon_{abc}\left|\right)$, where $G_{b}\left(a\right)$ is defined in Lemma 1, and $\rho_{t}:=\rho \left(Y^{*},Y_{t}\right)$.*


See Appendix D for proof.

To provide an intuition of Proposition 1, we visualize the flip probability in Figure 1. This flip probability provides a guideline for choosing the accepted sample size *c* given the datasize *N* and the desired privacy level $\epsilon_{total}$. For instance, if a given dataset is extremely small, e.g., containing datapoints on the order of 10, *c* has to be chosen such that the flip probability of each posterior sample remains low for a given privacy guarantee ($\epsilon_{total}$). If a higher number of posterior samples are needed, then one has to reduce the desired privacy level for the posterior sample of ABCDP to be similar to that of ABC. Otherwise, with a small $\epsilon_{total}$ with a large *c*, the accepted posterior samples will be poor. On the other hand, if the dataset is bigger, then a larger *c* can be taken for a reasonable level of privacy.

### 4.2. Convergence of Posterior Expectation of Rejection-ABCDP to Rejection-ABC

The flip probability studied in Section 4.1 only accounts for the effect of noise added to a single output of ABCDP. Building further on this result, we analyzed the discrepancy between the posterior expectations derived from ABCDP and from the rejection ABC. This analysis requires quantifying the effect of noise added to the whole sequence of outputs of ABCDP. The result is presented in Theorem 2.

**Theorem** **2.**
*Given $Y^{*}$ of size N, and ${\left\{\left(\theta_{t},Y_{t}\right)\right\}}_{t=1}^{T}$ as input, let ${\tilde{\tau }}_{t}\in \left\{0,1\right\}$ be the output from Algorithm 1 where ${\tilde{\tau }}_{t}=1$ indicates that $\left(\theta_{t},Y_{t}\right)$ is accepted, for t=1,…,T. Similarly, let $\tau_{t}$ denote the output from the traditional rejection ABC algorithm, for t=1,…,T. Let f be an arbitrary vector-valued function of **θ**. Assume that the numbers of accepted samples from Algorithm 1, and the traditional rejection ABC algorithm are $c:=\sum_{t=1}^{T}{\tilde{\tau }}_{t}\geq 1$ and $c^{\prime }:=\sum_{t=1}^{T}\tau_{t}\geq 1$, respectively. Let $b=\frac{4c{\sqrt{B}}_{k}}{\epsilon_{total}N}$ if RESAMPLE=True, and $b=\frac{2\left(c+1\right)\sqrt{B_{k}}}{\epsilon_{total}N}$ if RESAMPLE=False (see Theorem 1. Define $K_{T}:=\max_{t=1,\ldots ,T}{∥f\left(\theta_{t}\right)∥}_{2}$. Then, the following statements hold for both RESAMPLE options:*

*1. $E_{{\tilde{\tau }}_{1},\ldots ,{\tilde{\tau }}_{T}}∥\frac{1}{c}\sum_{t=1}^{T}f\left(\theta_{t}\right){\tilde{\tau }}_{t}-\frac{1}{c^{\prime }}\sum_{t=1}^{T}f\left(\theta_{t}\right)\tau {}_{t}{∥}_{2}\leq \frac{2K_{T}}{c^{\prime }}\sum_{t=1}^{T}G_{b}\left(\right|\rho_{t}-\epsilon_{abc}\left|\right),$ where the decreasing function $G_{b}\left(x\right)\in \left(0,\frac{1}{2}\right]$ for any x≥0 is defined in Lemma 2;*

*2. $E_{{\tilde{\tau }}_{1},\ldots ,{\tilde{\tau }}_{T}}∥\frac{1}{c}\sum_{t=1}^{T}f\left(\theta_{t}\right){\tilde{\tau }}_{t}-\frac{1}{c^{\prime }}\sum_{t=1}^{T}f\left(\theta_{t}\right)\tau {}_{t}{∥}_{2}\to 0$ as N→∞;*

*3. For any a>0:*
$$P\left(∥\frac{1}{c}\sum_{t=1}^{T}f\left(\theta_{t}\right){\tilde{\tau }}_{t}-\frac{1}{c^{\prime }}\sum_{t=1}^{T}f\left(\theta_{t}\right)\tau {}_{t}{∥}_{2}\leq a\right)\geq 1-\frac{4K_{T}}{3ac^{\prime }}\sum_{t=1}^{T}\exp\left(-\frac{|\rho_{t}-\epsilon_{abc}|}{2b}\right)$$
*where the probability is taken with respect to ${\tilde{\tau }}_{1},\ldots ,{\tilde{\tau }}_{T}$.*


Theorem 2 contains three statements. The first states that the expected error between the two posterior expectations of an arbitrary function *f* is bounded by a constant factor of the sum of the flip probability in each rejection/acceptance step. As we have seen in Section 4.1, the flip probability is determined by the scale parameter *b* of the Laplace distribution. Since b=O(1/N) (see Theorem 1 and Lemma 1), it follows that the expected error decays as *N* increases, giving the second statement.

The third statement gives a probabilistic bound on the error. The bound guarantees that the error decays exponentially in *N*. Our proof relies on establishing an upper bound on the error as a function of the total number of flips $\sum_{t=1}^{T}\left|{\tilde{\tau }}_{t}-\tau_{t}\right|$ which is a random variable. Bounding the error of interest then amounts to characterizing the tail behavior of this quantity. Observe that in Theorem 2, we consider ABCDP and rejection ABC with the same computational budget, i.e., the same total number of iterations *T* performed. However, the number of accepted samples may be different in each case (*c* for ABCDP and $c^{\prime }$ for reject ABC). The fact that *c* itself is a random quantity due to injected noise presents its own technical challenge in the proof. Our proof can be found in Appendix E.

## 5. Related Work

Combining DP with ABC is relatively novel. The only related work is [27], which states that a rejection ABC algorithm produces posterior samples from the exact posterior distribution given perturbed data, when the kernel and bandwidth of rejection ABC are chosen in line with the data perturbation mechanism. The focus of [27] is to identify the condition when the posterior becomes exact in terms of the kernel and bandwidth of the kernel through the lens of data perturbation. On the other hand, we use the sparse vector technique to reduce the total privacy loss. The resulting theoretical studies including the flip probability and the error bound on the posterior expectation are new.

## 6. Experiments

### 6.1. Toy Examples

We start by investigating the interplay between $\epsilon_{abc}$ and $\epsilon_{total}$, in a synthetic dataset where the ground truth parameters are known. Following [18], we also consider a symmetric Dirichlet prior π and a likelihood p(y|θ) given by a mixture of uniform distributions as
(11)$$\begin{array}{cc} \pi (\theta ) & =\mathrm{Dirichlet}(\theta ;1),\\ P(y|\theta ) & =\sum_{i=1}^{5}\theta_{i}\mathrm{Uniform}\left(y;\left[i-1,i\right]\right). \end{array}$$
A vector of mixing proportions is our model parameters θ, where the ground truth is $\theta^{*}={\left[0.25,0.04,0.33,0.04,0.34\right]}^{⊤}$ (see Figure 2). The goal is to estimate $E[\theta |Y^{*}]$ where $Y^{*}$ is generated with $\theta^{*}$.

We first generated 5000 samples for $Y^{*}$ drawn from (Equation 11) with true parameters $\theta^{*}$. Then, we tested our two ABCDP frameworks with varying $\epsilon_{abc}$ and $\epsilon_{total}$. In these experiments, we set $\rho =\hat{\mathrm{MMD}}$ with a Gaussian kernel. We set the bandwidth of the Gaussian kernel using the median heuristic computed on the simulated data (i.e., we did not use the real data for this, hence there is no privacy violation in this regard).

We drew 5000 pseudo-samples for $Y_{t}$ at each time. We tested various settings, as shown in Figure 3, where we vary the number of posterior samples, c={10,100,1000}, $\epsilon_{abc}=\left\{0.05,0.1,0.2,0.5\right\}$ and $\epsilon_{total}=\left\{0.5,1.0,10,\infty \right\}$. We showed the result of ABCDP for both RESAMPLE options in Figure 3.

### 6.2. Coronavirus Outbreak Data

In this experiment, we modelled coronavirus outbreak in the Netherlands using a polynomial model consisting of four parameters $a_{0},a_{1},a_{2},a_{3}$, which we aimed to infer, where:(12)$$\begin{array}{c} y\left(t\right)=a_{3}+a_{2}t+a_{1}t^{2}+a_{0}t^{3}. \end{array}$$

The observed (https://www.ecdc.europa.eu/en/publications-data/download-todays-data-geographic-distribution-COVID-19-cases-worldwide, accessed on 10 October 2020) data are the number of cases of the coronavirus outbreak from 27 February to 17 March 2020, which amounts to 18 datapoints (N=18). The presented experiment imposes privacy concern as each datapoint is a count of the individuals who are COVID positive at each time. The goal is to identify the approximate posterior distribution $\tilde{P}\left(a_{0},a_{1},a_{2},a_{3}|y^{*}\right)$ over these parameters, given a set of observations.

Recalling from Figure 1 that the small size of data worsens the privacy and accuracy trade-off, the inference is restricted to a small number of posterior samples (we chose c=5) since the number of datapoints is extremely limited in this dataset. We used the same prior distributions for the four parameters as $a_{i}∼N\left(0,1\right)$ for all i=0,1,2,3. We drew 50,000 samples from the Gaussian prior, and performed our ABCDP algorithm with $\epsilon_{total}=\left\{13,22,44\right\}$ and $\epsilon_{abc}=0.1$, as shown in Figure 4.

### 6.3. Modeling Tuberculosis (TB) Outbreak Using Stochastic Birth–Death Models

In this experiment, we used the stochastic birth–death models to model Tuberculosis (TB) outbreak. There are four parameters that we aim to infer, which go into the communicable disease outbreak simulator as inputs: burden rate β, transmission rate $t_{1}$, reproductive numbers $R_{1}$ and $R_{2}$. The goal is to identify the approximate posterior distribution $\tilde{p}\left(R_{1},t_{1},R_{2},\beta |y^{*}\right)$ over these parameters given a set of observations. Please refer to Section 3 in [28] for the description of the birth–death process of the model. We used the same prior distributions for the four parameters as in [28]: β∼N(200,30), $R_{1}∼Unif\left(1.01,20\right)$, $R_{2}∼Unif\left(0.01,\left(1-0.05R_{1}\right)/0.95\right)$, $t_{1}∼Unif\left(0.01,30\right)$.

To illustrate the privacy and accuracy trade-off, we first generated two sets of observations $y^{*}$ (n=100 and n=1000) by some *true* model parameters (shown as black bars in Figure 5). We then tested our ABCDP algorithm with a privacy level ϵ=1. We used the summary statistic described in Table 1 in [28] and used a weighted L2 distance as ρ as done in [28], together with $\epsilon_{abc}=150$. Since there is no bounded sensitivity in this case, we impose an artificial boundedness by clipping the distance by *C* (we set C=200) when the distance goes beyond *C*.

As an error metric, we computed the mean absolute distance between each posterior mean and the true parameter values. The top row in Figure 5 shows that the mean of the prior (red) is far from the true value (black) that we chose. As we increase the data size from n=100 (middle) to n=1000 (bottom), the distance between true values and estimates reduces, as reflected in the error from 4.71 to 2.20 for RESAMPLE = True; and from 4.51 to 2.10 for RESAMPLE=False.

## 7. Summary and Discussion

We presented the ABCDP algorithm by combining DP with ABC. Our method outputs differentially private binary indicators, yielding differentially private posterior samples. To analyze the proposed algorithm, we derived the probability of flip from the rejection ABC’s indicator to the ABCDP’s indicator, as well as the average error bound of the posterior expectation.

We showed experimental results that output a relatively small number of posterior samples. This is due to the fact that the cumulative privacy loss increases linearly with the number of posterior samples (i.e., *c*) that our algorithm outputs. For a large-sized dataset (i.e., *N* is large), one can still increase the number of posterior samples while providing a reasonable level of privacy guarantee. However, for a small-sized dataset (i.e., *N* is small), a more refined privacy composition (e.g., [29]) would be necessary to keep the cumulative privacy loss relatively small, at the expense of providing an *approximate* DP guarantee rather than the pure DP guarantee that ABCDP provides.

When we presented our work to the ABC community, we often received the question of whether we could apply ABCDP to other types of ABC algorithms such as the sequential Monte Carlo algorithm which outputs the significance of each proposal sample, as opposed to its acceptance or rejection as in the rejection ABC algorithm. Directly applying the current form of ABCDP to these algorithms is not possible, while applying the Gaussian mechanism to the significance of each proposal sample can guarantee differential privacy for the output of the sequential Monte Carlo algorithm. However, the cumulative privacy loss will be relatively large, as now it is a function of the number of proposal samples, whether they are taken as good posterior samples or not.

A natural by-product of ABCDP is differentially private synthetic data, as the simulator is a public tool that anybody can run and hence differentially private posterior samples suffice for differentially private synthetic data without any further privacy cost. Applying ABCDP to generate complex datasets is an intriguing future direction.

## Figures and Tables

**Figure 1 entropy-23-00961-f001:**
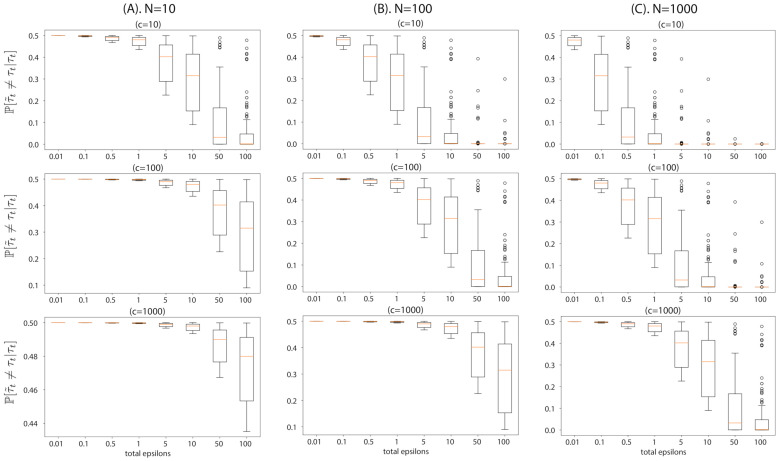
Visualization of flip probability derived in Proposition 1, the probability that the outputs of ABCDP and ABC differ given an output of ABC, with different dataset size *N* and accepted posterior sample size *c*. We simulated ρ∼Uniform[0,1] (drew 100 values for ρ) and used $\epsilon_{abc}=0.2$: (**A**) This column shows the flip probability at a regime of extremely small datasets, N=10. Top plot shows the probability at c=10, middle plot at c=100, and bottom plot at c=1000. In this regime, even $\epsilon_{total}=100$ cannot reduce the flip probability to perfectly zero when c=10. The flip probability remains high when we accept more samples, i.e., c=1000; (**B**) the flip probability at N=100; (**C**) the flip probability at N=1000. As we increase the dataset size *N* (moving from the left to right columns), the flip probability approaches zero at a smaller privacy loss $\epsilon_{total}$.

**Figure 2 entropy-23-00961-f002:**
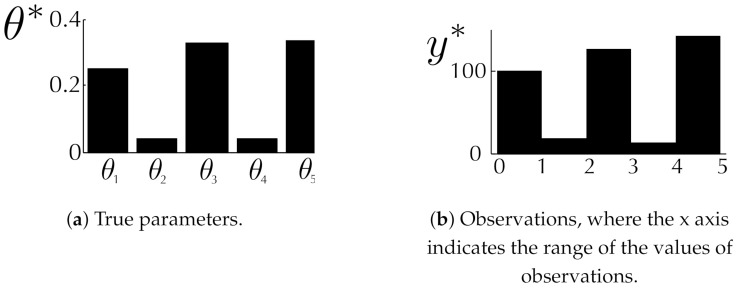
Synthetic data. (**a**): 5-dimensional true parameters; (**b**): observations sampled from the mixture of uniform distributions in (Equation 11) with the true parameters.

**Figure 3 entropy-23-00961-f003:**
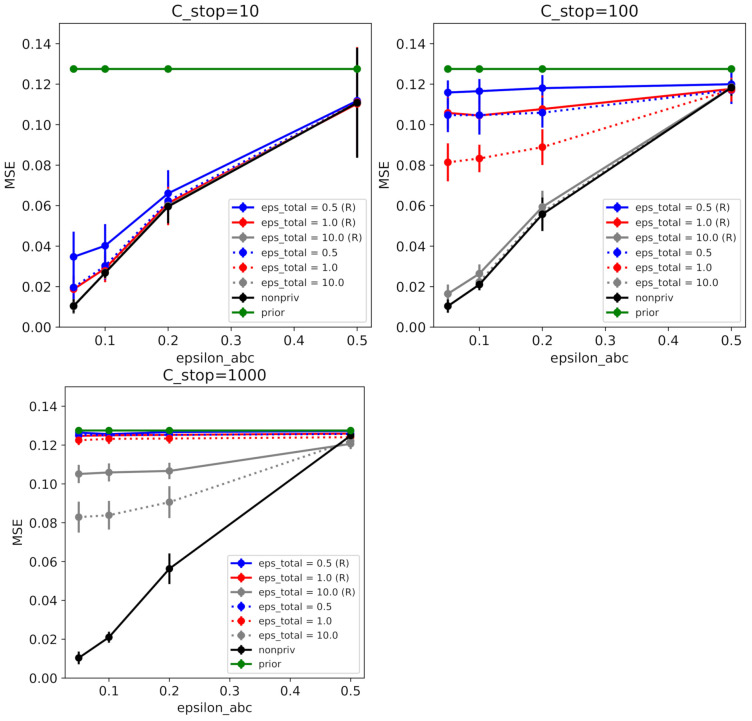
ABCDP on synthetic data. Mean-squared error (between true parameters and posterior mean) as a function of similarity threshold $\epsilon_{abc}$ given each privacy level. We ran ABCDP with the following options: *RESAMPLE = True* (denoted by R and solid line); or *RESAMPLE = False* (without R and dotted line) for 60 independent runs. (**Top Left**) When $c_{stop}=10$ at different values of $\epsilon_{abc}$, ABCDP and non-private ABC (black trace) achieved the highest accuracy (lowest MSE) at the smallest $\epsilon_{abc}$ ($\epsilon_{abc}=0.01$). Notice that ABCDP *RESAMPLE = False* (dotted) outperformed ABCDP *RESAMPLE=True* (solid) for the same privacy tolerance ($\epsilon_{total}$) at small values of $\epsilon_{abc}$. (**Top Right**) MSE for $c_{stop}=100$ at different values of $\epsilon_{abc}$; (**Bottom Left**) MSE for $c_{stop}=1000$ at different values of $\epsilon_{abc}$. We can observe when $\epsilon_{abc}$ is large, ABCDP (gray) marginally outperforms non-private ABC (black) due to the excessive noise added in ABCDP.

**Figure 4 entropy-23-00961-f004:**
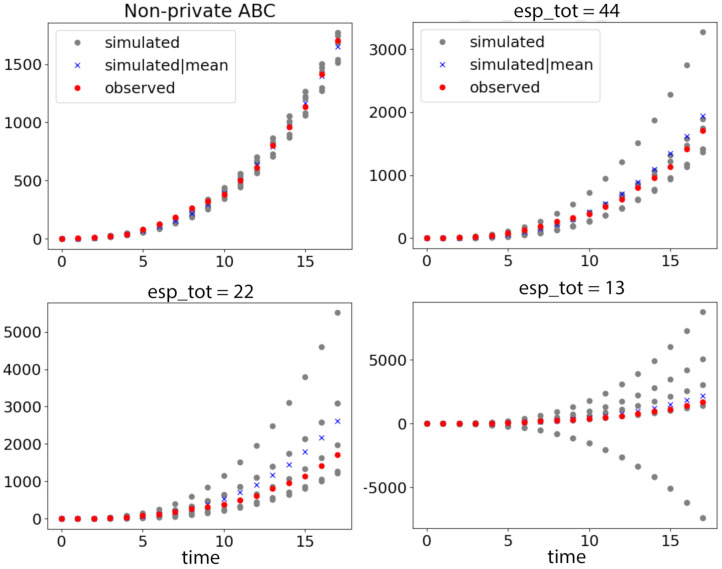
COVID-19 outbreak data (N=18) and simulated data under different privacy guarantees. Red dots show observed data, and gray dots show simulated data drawn from 5 posterior samples accepted in each case. The blue crosses are simulated data given the posterior mean in each case: (**Top left**) simulated data by non-private ABC; (**Top right**) simulated data by ABCDP with $\epsilon_{total}=44$ are relatively well aligned with regard to the extremely small size of the data. Note that we use a different scale for left and right plots for better visibility. If we use the same y scale in both plots, the simulated and observed points are not distinguishable on the left plot: (**Bottom left**) the simulated data given 5 posterior samples exhibit a large variance when $\epsilon_{total}=22$; and (**Bottom right**) when $\epsilon_{total}=13$, the simulated data exhibit an excessively large variance.

**Figure 5 entropy-23-00961-f005:**
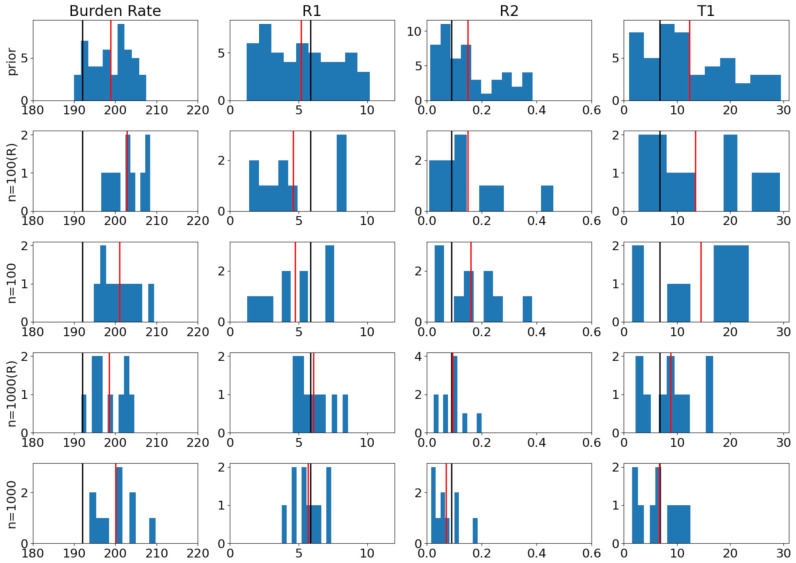
Posterior samples for modeling tuberculosis (TB) outbreak. In all ABCDP methods, we set $\epsilon_{total}=1$. True values in black. Mean of samples in red: (R) indicates ABCDP with Resampling = True. (**1st row**): Histogram of 50 samples drawn from the prior (we used the same prior as [28]); (**2nd row**): 10 posterior samples from ABCDP with (R) given n=100 observations; (**3rd row**): 10 posterior samples from ABCDP without (R) given n=100 observations; (**4th row**): 10 posterior samples from ABCDP with (R) given n=1000 observations; and (**5th row**): 10 posterior samples from ABCDP without (R) given n=1000 observations. The distance between the black bar (true) and red bar (estimate) reduces as the size of data increases from 100 to 1000. ABCDP with Resampling=False performs better regardless of the data size.

## Data Availability

A publicly available dataset was analyzed in this study. This data can be found here: https://www.ecdc.europa.eu/en/publications-data/download-todays-data-geographic-distribution-covid-19-cases-worldwide.

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
