# Peer review of "ABCDP: Approximate Bayesian Computation with Differential Privacy"

_entropy, 2021, doi:10.3390/e23080961_

Round 1
Reviewer 1 Report
The manuscript “ABCDP: Approximate Bayesian computation with differential privacy” describes a novel approximate Bayesian computation framework, which provides an algorithm for the generating of the non-private posterior datasets w.r.t the private prior dataset satisfying the difference privacy (DP) conditions.
The manuscript contains an introduction into the topic, description of the algorithm and proofs of the ε-DP and convergence of the algorithm. Numerical experiments illustrate the theoretical results.
In general, the manuscript provides a contribution a scientific relevant field of research. Nevertheless, the mathematical-logical consistency of the manuscript should be improved, before it could be considered for publication. My major concern is that the several assumptions, which are used in the proofs of the statements, are more restrictive, than the requirements provided in the claims of the theorems.
Detailed remarks.
- 98 and below: “MMD” or “MMD2”?
- Lemma 4.1 if the proof requires the Gaussian type kernel, than this restriction has to be a part of the statement of the lemma or at least a part of the general assumptions at the beginning of the manuscript, that clarifies, that the claims only valid for Gaussian/Squared exponential kernels.
- Theorem 4.1: The \epsilon-DP is not defined before.
- Algorithm 1 / Theorem 4.1 the joint use/choice of the problem parameters c,b,\delta_\rho should be explained at some position.
- Appendix: The use of the two different symbols P, \mathbf{P} and Pr for the probability measure is misleading.
- Appendix B: It should be explained, that is the feature map \phi of the kernel k. The claim of the Lemma don’t contains any assumptions on kernel. If kernel not supposed to be Gaussian anyway, than (depending of the definition of the kernel) the kernel has at least to be positive definite.
- In general, I miss the discussion / explanation of the theoretical results. It is hard to understand, the purpose of the several theorems / lemmas. It is also not clear which theorems are the key results. Furthermore, all required assumptions should be listed or referenced (it can also be an assumptions block in the beginning of the manuscript or “the same assumptions as in the Thm …”) in each Theorem / Lemma.
Author Response
- 98 and below: “MMD” or “MMD2”?
- MMD is generally used to mean "MMD squared". We will make it clear in the resubmission.
- Lemma 4.1 if the proof requires the Gaussian type kernel, than this restriction has to be a part of the statement of the lemma or at least a part of the general assumptions at the beginning of the manuscript, that clarifies, that the claims only valid for Gaussian/Squared exponential kernels.
- Lemma 4.1 is not restricted to a Gaussian type kernel. Lemma 4.1 and the supplied proof in Section B hold true for any positive definite kernel k that is uniformly bounded by B_k (which includes a Gaussian type kernel where B_k = 1 in that case, for instance). This boundedness assumption is stated as part of Lemma 4.1.
- Theorem 4.1: The \epsilon-DP is not defined before.
- It is clearly defined in Eq.(5) and the line below.
- Algorithm 1 / Theorem 4.1 the joint use/choice of the problem parameters c,b,\delta_\rho should be explained at some position.
- Eq.(8) defines the relationship between the three variables. Section 4.1 describes the interplay between them.
- Appendix: The use of the two different symbols P, \mathbf{P} and Pr for the probability measure is misleading.
- We will make it consistent.
- Appendix B: It should be explained, that is the feature map \phi of the kernel k. The claim of the Lemma don’t contains any assumptions on kernel. If kernel not supposed to be Gaussian anyway, than (depending of the definition of the kernel) the kernel has at least to be positive definite.
- \phi is the feature map of the kernel k, and the kernel k is positive definite throughout the paper. The discussion on the feature map \phi, positive definite kernels, and the MMD is given in Section 2 (background). Note that if k is positive definite, the Moore-Aronszajn theorem ensures that there exists a feature map (possibly infinite-dimensional) Ï• such that the inner product between Ï•(x) and Ï•(y) is k(x,y), for any x,y in the domain. We use this fact in the proof. We will elaborate more in Section 2.
- In general, I miss the discussion / explanation of the theoretical results. It is hard to understand, the purpose of the several theorems / lemmas. It is also not clear which theorems are the key results. Furthermore, all required assumptions should be listed or referenced (it can also be an assumptions block in the beginning of the manuscript or “the same assumptions as in the Thm …”) in each Theorem / Lemma.
- All lemmas/propositions/theorems already contain assumptions at the beginning of each of them (if any), as commonly done in the literature. For instance, in Theorem 4.1, assuming \rho has such a property as stated, then Algorithm 1 is epsilon-DP with the epsilon being Eq.(8). Theorems are the main results to pay attention to (and this is why they're called theorems). After presenting Theorem 4.3, we provided explanations on Theorem 4.1 and 4.3.
- We reiterate the purpose of each theorem/lemma/proposition here, although all of these are already in our first submission. The purpose of Theorem 4.1 is to give the formal relationship between the noise scale and the final privacy loss, which is written right before Theorem 4.1. The purpose of Lemma 4.2 is to compute the flip probability, we first compute the probability density function of the random variables m_t−ν_t, as written right before Lemma 4.2. The purpose of proposition 4.2. is to give a formal statement of the flip probability we're interested in, which is also written right before we presented proposition 4.2. The purpose of Theorem 4.3 is to analyze the discrepancy between the posterior expectations derived from ABCDP and from the rejection ABC. This analysis requires quantifying the effect of noise added to the whole sequence of outputs of ABCDP. These sentences are already included in our first submission.
Reviewer 2 Report
The authors introduced a rejection ABC variant using a sparse vector technique holding differentially private. In this sense, the rejection stage is coupled with a noise perturbation strategy within the ABC threshold and distance expressions.
Pros.
- Well-mathematical founded provided theoretical convergence and analysis of the proposal.
- Overall, the paper motivation and presentation are clear.
- Interesting framework for the ABC community.
Cons.
- The related work is limited. Though DP-based approaches are not typical for ABC, different rejection enhancement are provided in the literature. Please include and analyze the following references:
AKL-ABC: An Automatic Approximate Bayesian Computation Approach Based on Kernel Learning
Constructing summary statistics for approximate Bayesian computation: semi-automatic approximate Bayesian computation
Statistical inference for noisy nonlinear ecological dynamic systems.
Approximate Bayesian Computation with Indirect Summary Statistics
Approximate Bayesian Computation with Indirect Summary Statistics.
Kernel approximate Bayesian computation in population genetic inferences.
- I suggest including some of the relevant ABC rejection-based strategies (see suggested references and maybe sequential Monte Carlo) within some of the provided experiments to get an insight into the DP ABC benefits from both theoretical and experimental points of view.
- Line 160, you mentioned that a large T is suggested. Could you please extend this statement and discuss it concerning your experiments?
- The proposed approach is sensitive to two key hyperparameters: c and eps_tot. The c value is justified from Fig 1 experiments; nonetheless, the eps_tot seems to be tuned experimentally. Why choose your DP-ABC instead of the straightforward ABC rejection or sequential Monte Carlo if the user needs to fix more hyperparameters manually?
- Following the previous comment, could you please provide an idea concerning the DPABC computational cost?
Minor comments:
- Please enhance the literature overview in the introduction section and add the paper's agenda as the last paragraph.
- Briefly summarize the provided experiments in point 3 of your contributions (see line 80).
- Line 101, y_i set ~ Q?
- Figure 3 (b), xlabel?
- Figure 4, Could you please fix the scale variations? At least for the first row.
Author Response
- I suggest including some of the relevant ABC rejection-based strategies (see suggested references and maybe sequential Monte Carlo) within some of the provided experiments to get an insight into the DP ABC benefits from both theoretical and experimental points of view.
- This is a good suggestion. We will update the introduction accordingly in our revised submission.
- Line 160, you mentioned that a large T is suggested. Could you please extend this statement and discuss it concerning your experiments?
- Depending on the acceptance threshold, a necessary value of T varies to accept a c-number of posterior samples. A large T is generally recommended as there is no way to know when the ABCDP algorithm will release the c-number of posterior samples. We will elaborate on this in the revised manuscript.
- The proposed approach is sensitive to two key hyperparameters: c and eps_tot. The c value is justified from Fig 1 experiments; nonetheless, the eps_tot seems to be tuned experimentally. Why choose your DP-ABC instead of the straightforward ABC rejection or sequential Monte Carlo if the user needs to fix more hyperparameters manually?
- Note that eps_tot is a level of privacy that a user allows losing in running the rejection ABC algorithm. Once a user sets esp_tot to an allowed privacy loss -- not experimentally set -- this value limits the range of values b and c can take, due to Theorem 4.1. Or, conversely, one can choose b and c, and this results in a particular level of total privacy loss, eps_tot.
- Following the previous comment, could you please provide an idea concerning the DPABC computational cost?
- The computational cost of DP-ABC is the same as the computational cost of rejection ABC.
Thanks for the minor comments. We will update our revised manuscript accordingly.
Round 2
Reviewer 1 Report
The authors have addressed most of my remarks in appropriate form. However, it could be a part of the local community standards, but in my opinion mathematical statements like theorem, lemma etc. should provide sufficient information about used symbols. At least by references or naming.
Reviewer 2 Report
After reading the author comments, I suggest the paper for publication.
This manuscript is a resubmission of an earlier submission. The following is a list of the peer review reports and author responses from that submission.